# PPM1G dephosphorylates eIF4E in control of mRNA translation and cell proliferation

Peng Wang[1,3] , Zixian Li[2], Sung-Hoon Kim[1], Haijin Xu[4], Hao Huang[1], Chutong Yang[1], Abby Snape[1], Jung-Hyun Choi[1] , Sara Bermudez[1], Marie-Noelle Boivin[3] , Nicolas Ferry[3] , Jason Karamchandani[3], Bhushan Nagar[2] , Nahum Sonenberg[1] 

The mRNA 5′cap-binding eukaryotic translation initiation factor 4E (eIF4E) plays a critical role in the control of mRNA translation in health and disease. One mechanism of regulation of eIF4E activity is via phosphorylation of eIF4E by MNK kinases, which promotes the translation of a subset of mRNAs encoding pro-tumorigenic proteins. Work on eIF4E phosphatases has been paltry. Here, we show that PPM1G is the phosphatase that dephosphorylates eIF4E. We describe the eIF4E-binding motif in PPM1G that is similar to 4E-binding proteins (4E-BPs). We demonstrate that PPM1G inhibits cell proliferation by targeting phospho-eIF4E–dependent mRNA translation.

## Introduction

Canonical eukaryotic mRNA translation commences with the recognition of the mRNA 5′-m7GpppN (where m is a methyl group and N is any nucleotide) cap structure by the heterotrimeric eukaryotic translation initiation factor 4F (eIF4F) complex (Sonenberg & Gingras, 1998). eIF4F is comprised of a cap-binding subunit eIF4E, a large scaffolding protein eIF4G and an ATP-dependent RNA helicase eIF4A. eIF4G recruits the 43S pre-initiation complex to the mRNA, whereas eIF4A unwinds the secondary structure of the mRNA 5′UTR to facilitate pre-initiation complex loading and subsequent scanning until the initiation codon is engaged (Imataka & Sonenberg, 1997; Hinnebusch, 2014; Pelletier et al, 2015; Merrick & Pavitt, 2018; Pelletier & Sonenberg, 2019).

eIF4E is the least abundant translation initiation factor and thus plays a pivotal role in the control of translation initiation (Hiremath et al, 1985; Duncan et al, 1987). eIF4E can be sequestered from eIF4G by 4E-BPs (eIF4E-binding proteins) which constitute a family of three proteins (4E-BP1, 2 and 3) in mammals (Lin et al, 1994; Pause et al, 1994; Mader et al, 1995; Peter et al, 2015). In response to energy,

nutrition and growth factors, the mammalian mechanistic target of rapamycin complex1 (mTORC1) sequentially phosphorylates 4E-BP at multiple sites in a hierarchical order (T37, T46, T70, and S65) to cause the release of eIF4E, thereby promoting eIF4E-eIF4G association to reconstitute the eIF4F complex and subsequent cap-dependent mRNA translation initiation (Gingras et al, 1998, 1999).

eIF4E is phosphorylated by MAPK-interacting kinases 1 and 2 (MNKs) (Fukunaga & Hunter, 1997; Waskiewicz et al, 1997, 1999; Wang et al, 1998; Ueda et al, 2004). eIF4G serves as a docking site for MNKs which phosphorylate eIF4E at a single site, serine 209 (S209) (Wang et al, 1998; Waskiewicz et al, 1999; Ueda et al, 2004). MNK2 exhibits a high basal activity and maintains eIF4E phosphorylation under physiological conditions, whereas MNK1 has low basal activity and phosphorylates eIF4E in response to Ras pathway activation or under stress (Wang et al, 1998; Waskiewicz et al, 1999; Scheper et al, 2001; Ueda et al, 2004). The abolishment of eIF4E phosphorylation (Ser209A mutation) engendered a decrease in the translation of a subset of mRNA encoding the proteins MMP-3, MMP-9, SNAIL, Mcl1, Bcl-2, Cyclin D1, MYC, Ccl2, Ccl7, PD-L1, IκBα, and RANTES, which function in cell invasion, survival, oncogenesis, and inflammatory response (Nikolcheva et al, 2002; Wendel et al, 2007; Furic et al, 2010; Herdy et al, 2012; Martinez et al, 2015; Robichaud et al, 2015, 2018; Seidel et al, 2016; Ruan et al, 2020; Guo et al, 2021; Huang et al, 2021).

mTOR and MNKs activities are frequently up-regulated in cancer to promote the synthesis of pro-tumorigenic proteins by phosphorylating 4E-BPs and eIF4E, respectively. Thus, inhibiting their phosphorylation can potentially lead to cancer therapies (Silvera et al, 2010; Ruggero, 2013; Bhat et al, 2015; Pelletier et al, 2015; Siddiqui & Sonenberg, 2015). mTOR inhibitors (e.g., rapamycin analogs; rapalogs and TORKi; TOR kinase inhibitors) have shown only limited benefit in the clinic (Yu et al, 2022) and only one MNK inhibitor (Tomivosertib) is being tested in clinical trials for solid and hematological cancers (Jin et al, 2021).

PPM1G (protein phosphatase, Mg2+/Mn2+ dependent 1G; also called PP2Cγ) is a member of the metal-ion dependent serine/threonine protein phosphatase PPM (or PP2C) family. PPM1G is

[1]Department of Biochemistry and Goodman Cancer Institute, McGill University, Montreal, Canada   [2]Department of Biochemistry, Francesco Bellini Life Sciences Building, McGill University, Montreal, Canada   [3]Clinical Biological Imaging and Genetic (C-BIG) Repository, Montreal Neurological Institute-Hospital, Montreal, Canada   [4]Department of Physiology, McIntyre Medical Sciences Building, McGill University, Montreal, Canada

Correspondence: bhushan.nagar@mcgill.ca; nahum.sonenberg@mcgill.ca

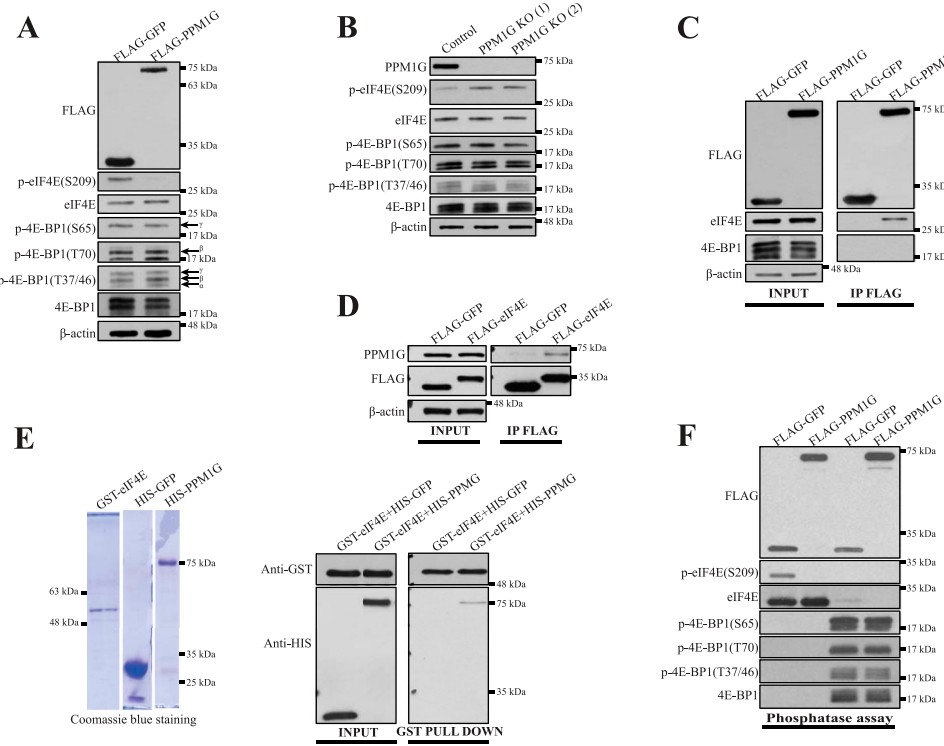

**Figure 1. PPM1G dephosphorylates eukaryotic translation initiation factor 4E (eIF4E) but not 4E-BP1.**
**(A)** PPM1G expression decreased eIF4E phosphorylation without affecting 4E-BP1 phosphorylation. HEK293H cells were transiently transfected with FLAG-GFP or FLAG-PPM1G. 24 h after the transfection, the cells were collected, and the cells lysates were analyzed by Western blotting (WB) for indicated proteins. The hyper-phosphorylated (β and γ) and hypo-phosphorylated (α) isoforms of 4E-BP1 were indicated. **(B)** PPM1G knockout increased eIF4E phosphorylation without affecting 4E-BP1 phosphorylation. A control and two PPM1G KO HEK293H cell lines were generated using Crispr-Cas9 coupled with gRNAs targeting a scramble sequence or different sites of PPM1G coding sequence, respectively, and their cells lysates were analyzed by WB for indicated proteins. **(C, D)** PPM1G associates with eIF4E. FLAG-tagged proteins transiently expressed in HEK293H cells were immunoprecipitated and their interactions with endogenous eIF4E, 4E-BP1, or PPM1G were determined by WB with the respective antibodies. **(E)** PPM1G directly binds to eIF4E. The interactions of bacterially purified GST-eIF4E (1 μg) with bacterially purified HIS-tagged proteins (100 ng) were tested using GST pull-down assay followed by the WB for indicated proteins (right panel). The quality of the recombinant proteins used in the GST pull-down assay was assessed by SDS/polyacrylamide gel electrophoresis and Coomassie blue staining (left panel). All HIS-tagged recombinant proteins were produced in *E. coli* strain BL21 and subsequently purified and eluted as described in the Materials and Methods section. Bacterially purified GST-eIF4E was purchased from Novus Biologicals. **(F)** PPM1G dephosphorylated eIF4E but not 4E-BP1 in an in vitro phosphatase assay. FLAG- and HA-tagged proteins transiently expressed in HEK293H cells were immunoprecipitated and subsequently eluted. The purified FLAG-GFP and FLAG-PPM1G were separately mixed with the purified HA-eIF4E or HA-4E-BP1 in the phosphatase buffer for 30 min (see the Materials and Methods section) and the phosphorylation and total level of HA-eIF4E and HA-4E-BP1 were analyzed by WB with the respective antibodies.
Source data are available for this figure.

distinct from other protein phosphatases in that it contains a long acidic region which is important for its binding to substrates and regulators (Travis & Welsh, 1997). PPM1G is essential for mouse embryonic viability (Foster et al, 2013) and regulates alternative splicing (Murray et al, 1999; Allemand et al, 2007; Chen et al, 2021), inflammatory responses, and cancer development (McNamara et al, 2013; Kumar et al, 2019; Hyder et al, 2020; Phannasil et al, 2020; Yu et al, 2020; Chen et al, 2021; Lin et al, 2021; Xiao et al, 2022; Xiong et al, 2022). PPM1G was reported to dephosphorylate 4E-BP1 (Liu et al, 2013), which is the predominant eIF4E-binding protein in most tissues, with the brain as a notable exception (Tsukiyama-Kohara et al, 2001). We therefore investigated whether PPM1G can suppress eIF4E-dependent mRNA translation by dephosphorylating 4E-BP1. Unexpectedly, we discovered that PPM1G dephosphorylates eIF4E, but not 4E-BP1, to control cell proliferation.

## Results

### PPM1G dephosphorylates eIF4E but not 4E-BP1

We studied the tumor suppressor activity of the phosphatase PPM1G as it was reported that PPM1G dephosphorylates 4E-BP1 (Liu et al,

2013). We expected that ectopic expression of PPM1G would reduce the phosphorylation of 4E-BP1 and prevent cap-dependent protein synthesis. Surprisingly, in HEK293H cells, PPM1G expression strongly diminished the phosphorylation of eIF4E but not the phosphorylation of 4E-BP1 (Fig 1A). Accordingly, the knockout (KO) of PPM1G promoted the phosphorylation of eIF4E without altering 4E-BP1 phosphorylation (Fig 1B). Notably, both PPM1G overexpression and KO did not affect MNKs' phosphorylation at sites (T197/202 for MNK1 and T334 for MNK2) which govern their activities (Ben-Levy et al, 1995; Fukunaga & Hunter, 1997; Meng et al, 2002) (Fig S1A and B), indicating that PPM1G directly dephosphorylates eIF4E. In support of this conclusion, immunoprecipitated FLAG-tagged PPM1G co-purified with endogenous eIF4E, but not 4E-BP1 (Fig 1C); and immunoprecipitated FLAG-tagged eIF4E associated with endogenous PPM1G (Fig 1D). However, the endogenous association of PPM1G with eIF4E is hardly detectable (~4 times stronger than background) (Fig S1C), suggesting a transient interaction. We next investigated whether PPM1G directly binds to eIF4E by using bacterially purified GST-tagged eIF4E to pull down bacterially purified HIS-tagged GFP and HIS-tagged PPM1G which were analyzed by SDS/polyacrylamide gel electrophoresis (Fig 1E, left figure). HIS-PPM1G, but not HIS-GFP, was pulled down with GST-eIF4E (Fig 1E, right figure), demonstrating that PPM1G binds to eIF4E. In addition, HA-tagged eIF4E, but not HA-tagged 4E-BP1, became dephosphorylated in an in vitro phosphatase assay (Fig 1F), further indicating that PPM1G directly

dephosphorylates eIF4E but not 4E-BP1. Moreover, the catalytically inactive form of PPM1G (PPM1G-D496A) (Murray et al, 1999) did not dephosphorylate eIF4E in an in vitro phosphatase assay (Fig S1D), indicating that PPM1G's activity is essential for eIF4E dephosphorylation in the test tube.

It was reported that the protein phosphatase 2A catalytic subunit (PP2A$_{C\alpha}$) dephosphorylates eIF4E (Li et al, 2010). However, overexpression of the PP2Ac$_\alpha$ suppressed the phosphorylation of eIF4E kinase MNK2 and its upstream kinase ERK activity (Fig S1E), whereas deletion of PP2Ac$_\alpha$ promoted their phosphorylation (Fig S1F). Moreover, eIF4E did not bind to PP2Ac$_\alpha$ (Fig S1G). Thus, PP2A$_{C\alpha}$ may promote eIF4E dephosphorylation indirectly. Taken together, our results indicate that PPM1G directly dephosphorylates eIF4E but not 4E-BP1, whereas PP2Ac$_\alpha$ may indirectly regulate eIF4E phosphorylation by targeting its upstream kinases.

### PPM1G contains a potential 4E-BM which mediates eIF4E binding

To delineate the region of PPM1G that interacts with eIF4E, we sought a potential 4E-binding motif (4E-BM) in the PPM1G sequence. eIF4E-binding proteins such as eIF4G, 4E-BPs, and 4E-T contain a canonical 4E-BM: YxxxxLΦ, where Y is tyrosine, L is leucine, x represents any amino acid and Φ is a hydrophobic residue (Mader et al, 1995; Marcotrigiano et al, 1999; Dostie et al, 2000; Romagnoli et al, 2021). PPM1G, and its orthologues, contain a conserved YxxxxL sequence but lack the Φ (Fig 2A), like the 4E-BM (LxxxxRS) in a human eIF4E-binding protein-CYFIP1 (Napoli et al, 2008) and the 4E-BM (YxxxxMK) in Drosophila 4E-BP (Bernal & Kimbrell, 2000). We therefore investigated whether the 4E-BM ($Y_{85}$xxxx$L_{90}$) is required for the interaction of PPM1G with eIF4E. Mutation of both $Y_{85}$ and $L_{90}$ residues of the 4E-BM into alanine (hereinafter referred to as YL$_{mut}$) caused a dramatic decrease in PPM1G's capacity (~75%) to dephosphorylate eIF4E and its binding (~80%) to eIF4E in HEK293H cells (Fig 2B). Moreover, bacterially purified HIS-PPM1G, but not HIS-PPM1G-YL$_{mut}$, was pulled down with GST-eIF4E, demonstrating that the 4E-BM is required for the direct binding of PPM1G to eIF4E (Fig S2). Although YL$_{mut}$ reduced PPM1G's binding to eIF4E, it did not affect PPM1G's association with its two established interacting proteins: B56$\delta$ and $\alpha$-catenin (Kumar et al, 2019) (Fig 2B), demonstrating that the 4E-BM of PPM1G may be specifically required for its interaction with eIF4E. We next examined whether the YL$_{mut}$ affects PPM1G's catalytic activity. PPM1G was reported to promote the dephosphorylation of $\alpha$-catenin in breast cancer cell lines (MCF7 and MDA-MB-231) (Kumar et al, 2019), which was not reproduced in HEK293H cells (Fig 2B). To address this discrepancy, FLAG-tagged GFP, PPM1G, and PPM1G-YL$_{mut}$ were purified from HEK293H cells (Fig 2C), and a phosphatase assay was performed in a test tube. PPM1G-YL$_{mut}$ retained about ~50% of the dephosphorylation activity of wild-type PPM1G toward eIF4E in vitro (Fig 2D) but did not affect $\alpha$-catenin phosphorylation (Fig 2E), supporting the conclusion that the YL$_{mut}$ does not abolish PPM1G activity.

### PPM1G shares eIF4E-binding residues with 4E-BP1

The residues $V_{69}$, $W_{73}$ and $L_{135}$ (VWL) of eIF4E mediate the eIF4E interaction with several binding partners: 4E-BP1, eIF4G and 4E-T (Siddiqui et al, 2012; Grüner et al, 2016; Romagnoli et al, 2021).

Mutation of the three residues to alanine in eIF4E (hereinafter referred to as VWL$_{mut}$) resulted in ~80% decrease in eIF4E expression (Fig 3A, left panel) most probably because of reduced stability (Murata & Shimotohno, 2006). We thus increased the input protein (6 mg) to saturate the anti-FLAG agarose-affinity gel (15 $\mu$l) in the immunoprecipitation experiment and obtained the immunoprecipitated FLAG-eIF4E and FLAG-eIF4E-VWL$_{mut}$ at a ratio of ~2:1 (Fig 3A, right panel). In contrast to FLAG-eIF4E, FLAG-eIF4E-VWL$_{mut}$ failed to associate with either PPM1G or 4E-BP1 (Fig 3A), demonstrating that the VWL residues are critical for the interaction of eIF4E with PPM1G and 4E-BP1. However, Angel1, which contains a canonical 4E-BM required for eIF4E binding (Gosselin et al, 2013), co-purified with both FLAG-eIF4E and FLAG-eIF4E-VWL$_{mut}$ (Fig 3A, right panel), indicating that the VWL residues of eIF4E are not required for all 4E-BM containing proteins binding. As both 4E-BP1 and PPM1G interact with the VWL residues of eIF4E, we investigated whether 4E-BP1 dephosphorylation and consequent binding to eIF4E interferes with eIF4E binding to PPM1G. FLAG-tagged PPM1G was purified from either mTOR inhibitor PP242-treated (Hoang et al, 2010) or untreated HEK293H cells. As expected, PP242 treatment led to the dephosphorylation of 4E-BP1 which disrupted ~90% PPM1G-eIF4E interaction without affecting the binding of PPM1G with B56$\delta$ and $\alpha$-catenin (Fig 3B), demonstrating that dephosphorylated 4E-BP1 may specifically preclude PPM1G-eIF4E interaction.

### PPM1G regulates mRNA translation and cell proliferation

We next investigated whether PPM1G interdicts mRNA translation. In a m7GDP (cap analog)-agarose pull-down assay experiment, eIF4G, but not PPM1G, was pulled down with m7GDP-bound eIF4E (Fig 4A). Moreover, as compared with GFP, PPM1G ectopic expression reduced (~60%) of the m7GDP-bound phospho-eIF4E (Fig 4A). Therefore, PPM1G most likely dephosphorylates cap-free eIF4E to reduce mRNA-bound phospho-eIF4E and the translation of a subset of mRNA. However, compared with GFP and PPM1G-YL$_{mut}$ ectopic expression, overexpression of PPM1G insignificantly reduced the polysomes-to-monosome ratio in HEK293H cells (Fig 4B), suggesting a nonessential role of PPM1G-phospho-eIF4E pathway in global mRNA translation. However, we expected that PPM1G would decrease phospho-eIF4E–dependent mRNA translation by suppressing eIF4E phosphorylation. Indeed, the expression of Mcl-1, a validated phospho-eIF4E target (Furic et al, 2010; Martinez et al, 2015; Robichaud et al, 2018), was significantly reduced (~30%) by PPM1G but not PPM1G-YL$_{mut}$ overexpression (Fig 4C).

We next examined the physiological consequence of the dephosphorylation of eIF4E by PPM1G. eIF4E phosphorylation promotes cell proliferation (Carter et al, 2016; D'Abronzo et al, 2017; Wheater et al, 2010). Thus, PPM1G is expected to impair cell proliferation by reducing eIF4E phosphorylation. In comparison with GFP expression, PPM1G ectopic expression caused a reduction in ~60% in the proliferation of HEK293H cells, whereas PPM1G-YL$_{mut}$ expression decreased by ~30% in cell proliferation (Fig 4D). PPM1G but not PPM1G-YL$_{mut}$ overexpression promoted the dephosphorylation of eIF4E in A549 (a non-small-cell lung cancer [NSCLC]) cells (Fig 4E) and impaired ~46% cell proliferation (Fig 4F). Taken together, our results show that PPM1G inhibits cell proliferation at least partly by dephosphorylating eIF4E.

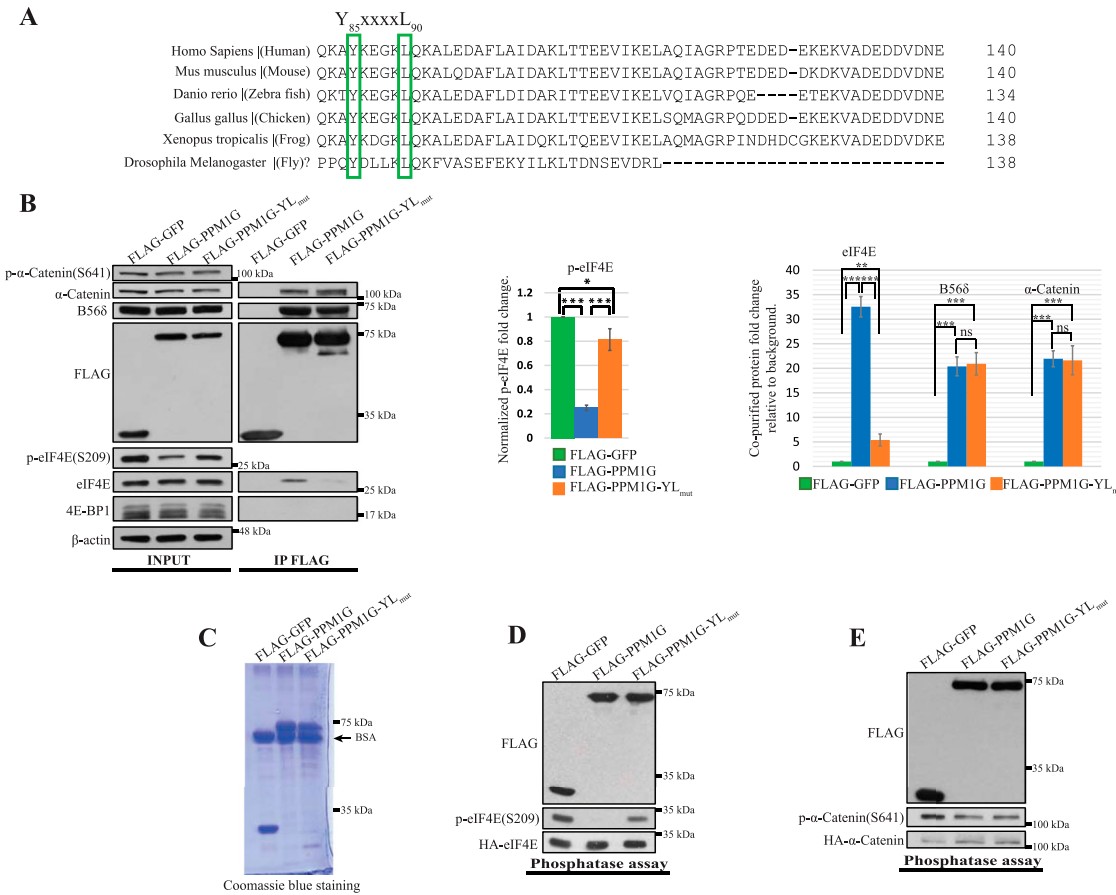

**Figure 2. The 4E-BM ($Y_{85}xxxxL_{90}$) motif of PPM1G is conserved in indicated PPM1G orthologues and required for eukaryotic translation initiation factor 4E (eIF4E) binding.**
**(A)** Indicated protein sequences of PPM1G orthologues from the NCBI gene database were aligned using CLUSTALO. The conserved Y and L residues in their 4E-BMs are highlighted with green frames. The question mark indicates that there is no typical PPM1G in drosophila and the selected phosphatase (CG10376, Flybase) is the sole drosophila PPM family member containing a specific 4E-BM which can be aligned with the 4E-BM ($Y_{85}xxxxL_{90}$) of human PPM1G. **(B)** The 4E-BM ($Y_{85}xxxxL_{90}$) of PPM1G is required for eIF4E but not B56$\delta$ and $\alpha$-catenin binding. FLAG-tagged proteins transiently expressed in HEK293H cells were immunoprecipitated and their interactions with indicated proteins were analyzed by WB (left figure, right panel) and quantitated (right column chart) with ImageJ. For quantification, the co-purified proteins were normalized to the immunoprecipitated FLAG-tagged proteins and the FLAG-GFP IP products were considered background. Values shown are the mean ± SE from three separate experiments. Asterisks indicate a statistically significant change determined by $t$ test ($^{NS}P > 0.05$, $*P < 0.05$, $**P < 0.01$, $***P < 0.001$). The middle column chart shows the relative fold change of phopho-eIF4E normalized to total eIF4E (left figure, left panel). **(C, D, E)** The quality of FLAG-tagged GFP, PPM1G, and PPM1G-YL$_{mut}$ used for the phosphatase assay in (D, E) were assessed by SDS/polyacrylamide gel electrophoresis and Coomassie blue staining. FLAG-tagged proteins transiently expressed in HEK293H cells were immunoprecipitated using anti-FLAG affinity gel and subsequently eluted with FLAG peptides in phosphatase buffer. **(D, E)** The 4E-BM ($Y_{85}xxxxL_{90}$) is required for PPM1G to efficiently dephosphorylate eIF4E (D) but not $\alpha$-catenin (E) in phosphatase assays. The purified FLAG-GFP, FLAG-PPM1G, and PPM1G-YL$_{mut}$ were separately mixed with the purified HA-eIF4E or HA-$\alpha$-catenin in the phosphatase buffer for 30 min (see the Materials and Methods section) and then the phosphorylation and total level of HA-eIF4E and HA-$\alpha$-catenin were analyzed by WB with the respective antibodies.
Source data are available for this figure.

# Discussion

Phosphorylation of eIF4E is linked to cancer development and progression (Yang et al, 2020). MNKs-eIF4E phosphorylation targeted therapy is in clinical trials (Xie et al, 2019). Therefore, it is necessary to understand how eIF4E phosphorylation is controlled by phosphatases.

It was reported that PPM1G dephosphorylates 4E-BP1 (Liu et al, 2013) but whether it dephosphorylates eIF4E was not examined in the latter study. We demonstrated that PPM1G associates with and dephosphorylates eIF4E but not 4E-BP1 (Fig 1A–C and F).

Although this discordance needs further investigation, both studies demonstrated that PPM1G regulates eIF4E-dependent mRNA translation. Indeed, compared with GFP overexpression, PPM1G overexpression decreased (~30%) the average value of polysomes-to-monosome ratio (Fig 4B) and reduced ~30% expression of Mcl-1 (Fig 4C), which is a validated phospho-eIF4E target (Wendel et al, 2007; Furic et al, 2010; Martinez et al, 2015).

The catalytic subunit of protein phosphatase 2A (PP2A$_{C\alpha}$) was reported to dephosphorylate both 4E-BP1 (Nanahoshi et al, 1998; Bishop et al, 2006; Guan et al, 2007) and eIF4E (Li et al, 2010). However, PP2A$_{C\alpha}$ is involved in a series of signaling pathways, such

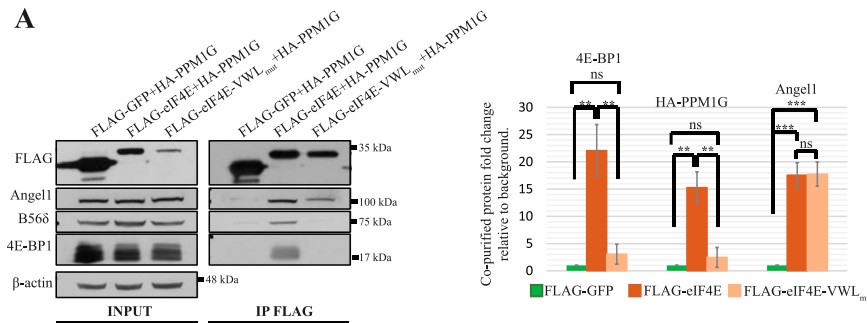

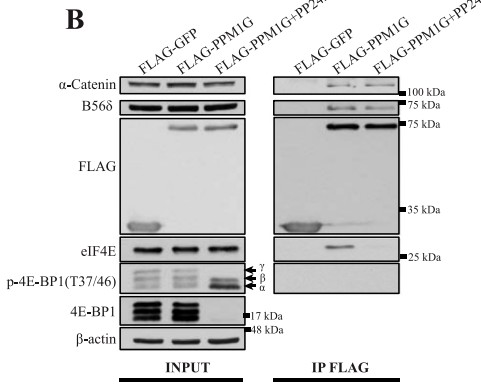

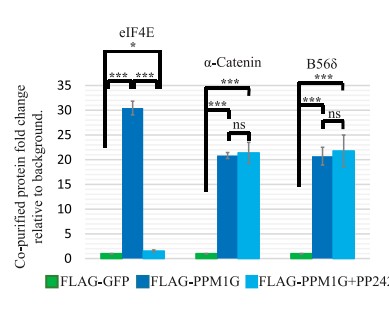

**Figure 3. PPM1G shares eukaryotic translation initiation factor 4E (eIF4E) binding residues with 4E-BP1.**

**(A)** eIF4E residues V69, W73, and L135 (VWL) are required for PPM1G and 4E-BP1 but not Angel1 binding. FLAG-tagged proteins transiently co-expressed with HA-PPM1G in HEK293H cells were immunoprecipitated and their interactions with indicated proteins were analyzed by WB (left figure) and then quantitated (right column chart). For quantification, the co-purified proteins were normalized to the immunoprecipitated FLAG-tagged proteins (left figure, right panel) and the FLAG-GFP IP products are considered background. Values shown are the mean ± SE from three separate experiments. Asterisks indicate a statistically significant change determined by $t$ test ($^{NS}P > 0.05$, $*P < 0.05$, $**P < 0.01$, $***P < 0.001$). **(B)** Dephosphorylated 4E-BP1 precludes PPM1G-eIF4E but not PPM1G-B56$\delta$/$\alpha$-catenin interaction. HEK293H cells were transiently transfected with FLAG-GFP or FLAG-PPM1G. 24 h after the transfection, the cells were treated with DMSO or mTOR inhibitor PP242 for 2 h. **(A)** FLAG-tagged proteins were subsequently immunoprecipitated from these cells and their interactions with indicated proteins were analyzed by WB and then quantitated as described in (A). The hyper-phosphorylated ($\beta$ and $\gamma$) and hypo-phosphorylated ($\alpha$) isoforms of 4E-BP1 were indicated.

Source data are available for this figure.

as MEK, ERK, AKT, mTORC1, TGF$\beta$, and Wnt (Westermarck et al, 2001; Yu et al, 2004; Yokoyama & Malbon, 2007; Wang et al, 2010; Apostolidis et al, 2016; Xiong et al, 2019; Xiao et al, 2020), among which ERK and mTORC1 are the established eIF4E and 4E-BP1 upstream kinases, respectively. Indeed, we showed that PP2A$_{C\alpha}$ but not PPM1G dramatically affects eIF4E upstream kinase MNK2 and MNK2 upstream kinase ERK activity (Fig S1A, B, E, and F). Moreover, eIF4E does not associate with PP2A$_{C\alpha}$ (Fig S1G). Therefore, PP2A is probably an indirect repressor of eIF4E phosphorylation.

We showed that the 4E-BM, located in the beginning of the long acidic region of PPM1G, contains the $Y_{85}xxxxL_{90}$ consensus sequence but with a glutamine ($Q_{91}$) in place of $\Phi$ (a hydrophobic amino acid) following leucine 90 (Fig 2A). The 4E-BM (YxxxxMK) of drosophila 4E-BP does not possess a $\Phi$ either and contains instead a lysine (K) which has an aliphatic side chain (Bernal & Kimbrell, 2000). The $\Phi$ is also absent in the 4E-BM (LxxxxRS) of CYFIP1, a human eIF4E-binding protein (Napoli et al, 2008). We further showed that $Y_{85}$;$L_{90}$ residues of the 4E-BM are required for PPM1G to interact with eIF4E and efficiently dephosphorylate eIF4E in cells and test tubes (Fig 2B and D). Notably, the 4E-BM is required neither for PPM1G activity (Fig 2E) nor for the binding of two other PPM1G partners: B56$\delta$ and $\alpha$-catenin (Kumar et al, 2019) (Fig 2B), suggesting that it may function specifically for eIF4E binding.

eIF4E residues V69, W73 and L135 that are required for 4E-BP1 binding (Siddiqui et al, 2012; Grüner et al, 2016) are also essential for PPM1G binding (Fig 3A). Indeed, dephosphorylated 4E-BP1

prevented PPM1G-eIF4E binding but not PPM1G-B56$\delta$ and PPM1G-$\alpha$-catenin binding (Fig 3B), probably because both 4E-BP1 and eIF4G also contain an additional non-canonical 4E-BM (Igreja et al, 2014; Grüner et al, 2016) which is absent in PPM1G, imparting higher affinity for eIF4E.

We showed that PPM1G does not associate with cap-bound eIF4E (Fig 4A) because eIF4E can only bind to the cap with strong affinity by associating with eIF4G (Haghighat et al, 1995; von Der Haar et al, 2000). Therefore, PPM1G most likely dephosphorylate cap-free eIF4E to reduce cap-bound phospho-eIF4E and the translation of a subset of mRNA including Mcl-1 (Fig 4C) but not the overall protein synthesis (Fig 4B). We further demonstrated that PPM1G inhibits the proliferation of both HEK293H cells and A549 cells (Fig 4D and F) partly by suppressing eIF4E phosphorylation, suggesting a conserved function.

In conclusion, our study documents a detailed molecular mechanism of eIF4E dephosphorylation by PPM1G and reveals the importance of PPM1G-phospho-eIF4E pathway in the restriction of cell proliferation. We wish that our study will help to develop a better eIF4E phosphorylation-targeted therapeutic strategy.

## Limitations of the study

It is imperative to obtain an atomic resolution of the PPM1G-eIF4E complex. The study of dephosphorylation kinetic of eIF4E by PPM1G and further investigations of the association/dissociation kinetics of eIF4E with PPM1G, MNKs, 4E-BPs, and eIF4G under physiological

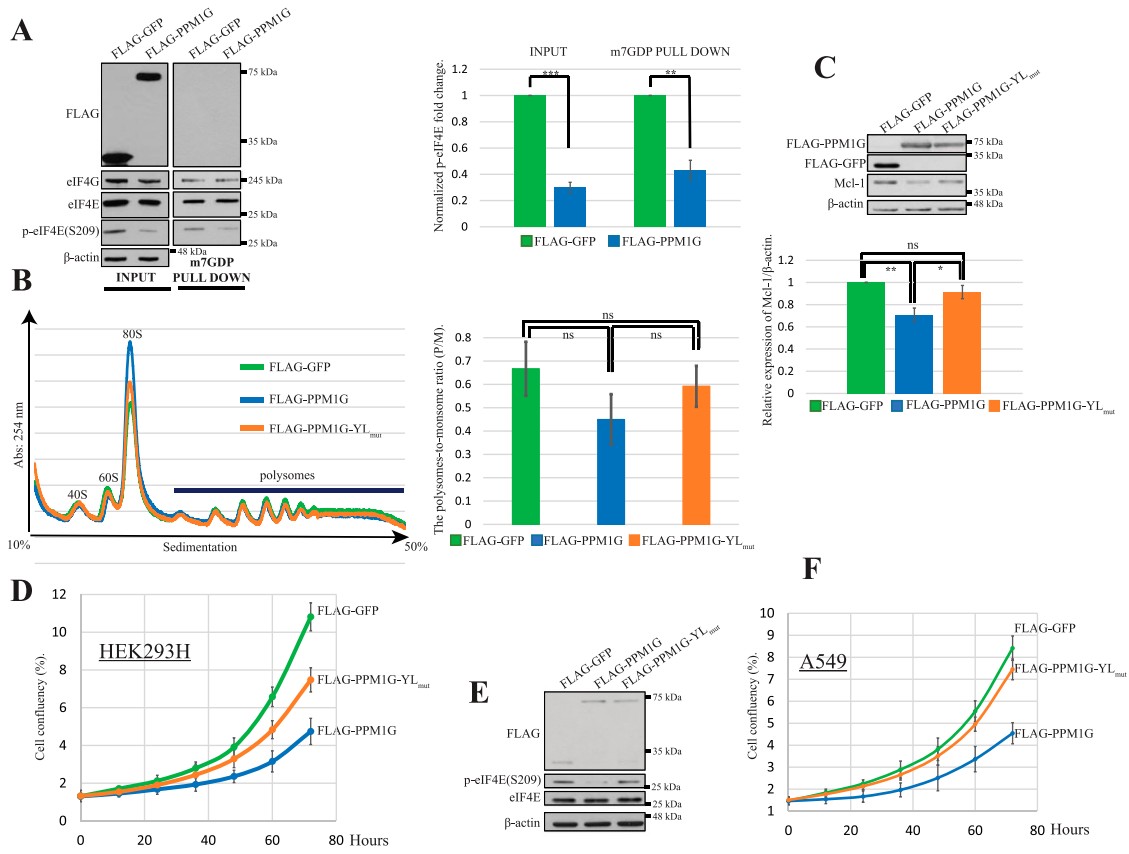

**Figure 4. PPM1G regulates p-eIF4E-dependent mRNA translation.**

**(A)** PPM1G reduced cap-bound p-eIF4E. HEK293H cells were transfected with FLAG-GFP or FLAG-PPM1G. 24 h after the transfection, cells were collected, and the cell lysates were incubated with m7GDP (cap analog) agarose to pull down the cap-associated protein complex. Indicated proteins and their bindings with m7GDP were analyzed by WB (left figure) and quantitated with ImageJ. The right column chart shows the relative fold change of p-eIF4E normalized to total eIF4E. Values shown are the mean ± SE from three separate experiments. Asterisks indicate a statistically significant change determined by $t$ test (*$P < 0.05$, **$P < 0.01$, ***$P < 0.001$). **(B)** Polysome profile analyses of HEK293H cells transiently expressing indicated FLAG-tagged proteins were analyzed by sucrose gradient sedimentation. For the calculation of the polysomes/monosome ratio (the right column chart), the areas under 80S peak and polysomes peaks were measured with ImageJ for each polysome profile graph and the area of the first seven polysomes peaks was divided by the area of peak for the 80S monosome. Values shown are the mean ± SE from three separate experiments and "ns" indicates no statistically significant change determined by $t$ test ($P > 0.05$). **(C)** PPM1G expression reduced Mcl-1 expression. HEK293H cells were transiently transfected with FLAG-GFP or FLAG-PPM1G or FLAG-PPM1G-YL$_{mut}$. 24 h after the transfection, cells were lysed, and the indicated proteins were analyzed by WB (upper figure) and quantitated with ImageJ (the lower column chart). Values shown are the mean ± SE from three separate experiments. Asterisks indicate a statistically significant change determined by $t$ test ($^{NS}P > 0.05$, *$P < 0.05$, **$P < 0.01$, ***$P < 0.001$). **(D, F)** HEK293H and A549 cells were transiently transfected with FLAG-GFP or FLAG-PPM1G or FLAG-PPM1G-YL$_{mut}$. 24 h after the transfection, cells were resuspended, and the same number of living cells was spread into a six-well plate for cell proliferation monitoring with the IncuCyte Zoom system. **(E)** PPM1G but not PPM1G-YL$_{mut}$ expression caused the dephosphorylation of eIF4E in non-small cell lung cancer A549 cells. The phosphorylation and total level of eIF4E in the lysates of A549 cells transiently expressing indicated FLAG-tagged proteins were analyzed by WB.
Source data are available for this figure.

and pathological conditions are needed for a full understanding of the regulatory mechanism of eIF4E phosphorylation.

# Materials and Methods

### Mutagenesis and plasmids

DNA mutations were generated with the Quick Change Lightning Site-Directed Mutagenesis Kit (Cat#21508; Agilent) according to the manufacturer's instructions. Plasmids pHA-eIF4E (Cat#17343) and pLN-HA-4E-BP1 (Cat#79438) were from Addgene. The pCMV3-FLAG-PPM1G (Cat#HG-11245-NF) was from Sino Biological Inc. and

pcDNA3-FLAG-PP2A$_{C\alpha}$ was a gift from Anne-Claude Gingras. Other mammalian cell and bacteria expression vectors were generated in the Gateway system (Invitrogen). The pDONR221λ or 223 entry vectors containing the coding sequence of the gene of interest were recombined with the expression vector to generate the following plasmids: pDEST-527-6xHIS-PPM1G, pDEST-527-6xHIS-GFP, pcDNA3-FLAG-GFP, pcDNA3-FLAG-eIF4E, pcDNA3-FLAG-eIF4E$_{Capmut}$, pcDNA3-FLAG-eIF4E$_{VWLmut}$, pMH-HA-GFP, pMH-HA-PPM1G, pMH-HA-$\alpha$-catenin.

### Cells

The HEK293H cell line was a gift from Joseph Marcotrigiano, the HEK293T cell line was from Thermo Fisher Scientific and the A549

cell line was provided by Sidong Huang. Cell lines were maintained in complete DMEM media with 10% FBS and 100 units/ml penicillin-streptomycin, at 37°C with 5% CO$_2$ and were confirmed to be mycoplasma-free by PlasmoTest (#rep-pt1; Invivogen).

## Transient transfection

$2 \times 10^6$ cells were seeded in a complete medium in a 10 cm dish for overnight growth. 30 µl Lipofectamine 2000 and 5–10 µg DNA (5 µg for HEK293H cells and 10 µg for A549 cells) were separately diluted in 250 µl Opti-MEM medium. After 5 min DNA and Lipofectamine were combined for 15 min at RT and added to cells in a dropwise manner. The cell medium was replaced by OPTI-MEM before the transfection. After the addition of the Lipofectamine-DNA mixture, cells were maintained at 37°C for 6 h in a CO$_2$ incubator. The OPTI-MEM was subsequently replaced by 10 ml fresh medium and 24 h later, cells were collected for Western blot analysis or treated for other experiments.

## Generation of cell lines

For the PPM1G knockout cell lines, 5 µg of pCLIP-Cas9-hCMV-tRFP and 5 µg of pCLIP-Dual-SFFV-ZsGreen containing two Cas9 guide RNAs (sgRNA) targeting PPM1G, or a scrambled sequence were co-transfected into HEK293H cells. 48 h after transfection, cells were resuspended and sequentially diluted to one cell per 100 µl of medium. Cells were transferred into a 96-well plate with 100 µl medium (one cell) per well. Upon reaching confluency, cells were transferred into a 48-well plate. Cells that lost expression of PPM1G, GFP (sgRNA vector), and RFP (Cas9 vector) were selected for further experiments.

For generation of stable cell lines, 7 µg of DNA or shRNA with 7 µg of each lentiviral packaging plasmid (PLP1, PLP2, and VSVG) were co-transfected into HEK293T cells using 50 µl of Lipofectamine 2000. 48 h after transfection, cell culture medium was collected and passed through a 0.45 µm filter. $1 \times 10^6$ HEK293H cells were incubated with the lentivirus-containing medium supplied with 5 mg/ml polybrene. After 48 h, virus-containing medium was discarded, and cells were cultured in medium supplied with selection antibiotic (4 µg/ml puromycin for shRNA expressing cells).

## Immunoblotting, immunoprecipitation, and m$^7$GDP pull-down

Cells were resuspended using a cell scraper. After centrifugation at 300$g$ for 5 min at 4°C, the cell medium was removed, and cells were resuspended in 1x ice-cold PBS. PBS was discarded after a centrifugation at 300$g$ for 5 min at 4°C and ice-cold lysis buffer (50 mM HEPES-KOH pH 7.5, 50 mM KCl, 2 mM MgCl$_2$, 2 mM EGTA, 5% glycerol, 0.5% NP40, and EDTA-free protease inhibitor cocktail [Roche] in double-distilled water) was added. Next, cell lysates were centrifuged at 15,000 rpm (rotor: Eppendor FA24x2) for 5 min at 4°C and the supernatant was collected. Total protein concentration was determined using the Bio-Rad Protein Assay Dye Reagent Concentrate (Cat#5000006). For immunoblotting, collected supernatant was mixed with an equal volume of 2x Laemmli sample buffer (Cat# 1610737; Bio-Rad) and heated for 5 min at 95°C. After centrifugation, the sample (30 µg) was resolved on a SDS–PAGE gel. Proteins were transferred to a nitrocellulose membrane, blocked

with 5% milk in TBST (TBS + 0.1%Tween), washed with TBST and incubated overnight with antibodies in 5% BSA in TBST on a shaker at 4°C. The next day, the membrane was washed with TBST, then incubated with HRP (horseradish peroxidase) conjugated secondary antibodies in 5% milk in TBST for 1 h at RT. After three washes with TBST, the membrane was incubated with ECL (Enhanced Chemiluminescence) for 1 min and exposed against an X-ray film. Antibodies were removed with stripping buffer (2 mM glycine pH 2.0, 0.1% SDS) for re-probing with different antibodies. For immunoprecipitation, 1–2 mg of total protein was incubated with pre-washed 25 µl of settled anti-FLAG affinity gel or 25 µl of anti-HA magnetic beads in a tube rotator at 4°C for 1 h. After a 5-min centrifugation at 2,000 rpm (rotor: Eppendorf FA24x2) at 4°C, the supernatant was removed, and the gel or beads were washed three times with cell lysis buffer. Proteins were eluted from the gel or beads by adding 30 µl of 2x Laemmli buffer (Bio-Rad). If not otherwise indicated, 20 µg (~2%) input and 30–50% IP products were used for the WB analysis of protein immunoprecipitation in this study. For m7GDP pull-down assay, 0.5 mg of total protein was incubated with pre-washed 15 µl of settled m7GDP-agarose beads (Jena Bioscience) in a tube rotator for 30 min at 4°C. Supernatant was removed after a 5-min centrifugation at 2,000 rpm (rotor: Eppendorf FA24x2) at 4°C. After three washes with cell lysis buffer, proteins were eluted by adding 30 µl of 2x Laemmli buffer and 10 µl pull-down products was used for WB analysis.

## Phosphatase assay

FLAG-GFP and FLAG-PPM1G were immunoprecipitated from HEK293H cells using 30 µl of settled anti-FLAG affinity gel. Proteins were eluted three times with 30 µl of 0.1 mg/ml FLAG peptide dissolved in phosphatase assay buffer (25 mM Tris–HCl, pH 7.5, 1 mM EDTA, 1 mM EGTA, 1 mM DTT, and 0.25 mg/ml BSA) (Adams & Wadzinski, 2007). HA-tagged 4E-BP1, eIF4E, and α-catenin proteins were purified from the HEK293H cells using 20 µl of settled anti-HA magnetic beads and eluted three times with 30 µl of 1 mg/ml HA peptide dissolved in phosphatase assay buffer. Elution of each protein was combined into an Eppendorf tube and protein concentration and purity were estimated using SDS/Polyacrylamide gel electrophoresis and Coomassie blue staining. Eluted proteins were stored at –80°C. For the phosphatase assay, 500 ng of FLAG-GFP and FLAG-PPM1G were separately incubated with 50 ng of HA-tagged 4E-BP1, eIF4E, or α-catenin in phosphatase buffer containing 2 mM MnCl$_2$ (Liu et al, 2013) (45 µl total volume) in an Eppendorf ThermoMixer (30°C, 1,000 rpm) for 30 min. The reaction was stopped by mixing with 15 µl of 4x Laemmli sample buffer (Bio-Rad) and heating at 95°C for 5 min, and proteins (10 µl) were analyzed by Western blot using total and phosphorylation site-specific antibodies.

## Recombinant protein production and GST-pull-down assay

BL21 competent cells were transformed with 100 ng of pDEST-527-6xHIS-GFP or PPM1G and spread on Lysogeny agar plates containing 100 µg/ml ampicillin. After overnight incubation at 37°C, a single colony was picked and incubated in 50 ml Lysogeny Broth (LB) buffer containing 100 µg/ml ampicillin for overnight in a

bacterial incubator. The day after, 20 ml of bacteria in LB were transferred to 1,000 ml of fresh LB buffer with ampicillin and grown in a bacterial incubator until at 600 nm OD 0.4–0.6 was reached. IPTG (isopropyl $\beta$-d-1-thiogalactopyranoside) was added with a final concentration of 1 mM and the bacteria were cultured overnight at 18°C. The next day, bacteria were collected by centrifugation, lysed in 20 ml lysis buffer (20 mM Tris–HCl pH 8, 500 mM NaCl, 10 mM imidazole, 10% glycerol, 0.1% Triton, and 0.1% sodium lauroyl sarcosinate), sonicated, and centrifuged. The supernatant was retained, and 0.5 ml of settled Ni-NTA resin was added and incubated on a wheel at 4°C for 1 h. After three washes with 1x PBS containing 25 mM imidazole, proteins were eluted with 250 mM imidazole in 1x PBS. Protein (5 $\mu$l) concentration and purity were analyzed using SDS/Polyacrylamide gel electrophoresis and Coomassie blue staining. For the GST pull-down assay, 1 $\mu$g of GST-eIF4E (Cat# LS-G39814; LSBio) was incubated with 0.5 $\mu$l of settled glutathione magnetic agarose beads (Cat# 78601; Thermo Fisher Scientific) in equilibration buffer (according to manufacturer's instructions) on a rotator at 4°C for 30 min. After three washes with equilibration buffer, GST-eIF4E was separately incubated with 100 ng of 6xHIS-tagged GFP and PPM1G in equilibration buffer on a rotator at 4°C for 30 min. Beads were washed three times with equilibration buffer and the proteins were eluted by adding 30 $\mu$l of 2x Laemmli buffer (Bio-Rad) and 10 $\mu$l was used for the WB analysis.

### IncuCyte analysis

24 h after transfection, cells were resuspended with trypsin and viable cell number was determined using a cell counter (Bio-Rad) by excluding Trypan blue-stained dead cells. $0.1 \times 10^6$ cells were seeded in a six-well plate. Once attached to the bottom, cell proliferation was monitored using the IncuCyte ZOOM system (Essen BioScience).

### Polysome profiling

Polysome profiling was performed according to a published protocol (Gandin et al, 2014). HEK293H cells were transfected with FLAG-GFP, FLAG-PPM1G, and FLAG-PPM1G-YL$_{mut}$ in a 15 cm plate. 24 h later, cells (90% confluency) were treated with 100 $\mu$g/ml cycloheximide (CHX) for 5 min at 37°C. Cells were washed twice with ice-cold PBS containing CHX and collected by scraping. Cells were centrifuged at 1,000 rpm using an FA24x2 rotor (Eppendorf) for 5 min at 4°C and lysed in hypotonic buffer (5 mM Tris–HCl, pH 7.4, 1.5 mM KCl, 2.5 mM MgCl$_2$, 200 U/ml RNase inhibitor, 2 mM DTT, 1 x protease inhibitor, 100 $\mu$g/ml CHX, 0.5% Triton X-100, and 0.5% sodium deoxycholate). After centrifugation at 15,000 rpm for 5 min at 4°C, the supernatant was retained, and the RNA concentration was determined using a NanoDrop 2000 (Thermo Fisher Scientific). Equal amounts of each sample were loaded onto a 10–50% sucrose gradient and then centrifuged at 36,000 rpm (rotor: SW 40 Ti) for 2 h at 4°C in an Optima L-80 XP ultracentrifuge (Beckman). Polysome fractions were collected using a Teledyne ISCO fractionator, and the optical density at 254 nm was measured using TracerDAQ.

All our results were reproduced at least three times. If not otherwise indicated, 30 $\mu$g total protein was loaded on the gel for the WB analysis.

## Data Availability

Original data are provided, and plasmids generated in this study are obtainable upon request.

## Supplementary Information

## Acknowledgements

This work was supported by Cancer Research Society (CRS) operating grant from 2020 to 2022. We thank professor Gergely Lukacs at McGill University and Professor Vincent Archambault at University of Montreal for their instructive advice and suggestions.

### Author Contributions

P Wang: conceptualization, data curation, formal analysis, funding acquisition, validation, investigation, methodology, project administration, and writing—original draft, review, and editing.
Z Li: Investigation.
S-H Kim: investigation.
H Xu: investigation.
H Huang: investigation.
C Yang: investigation.
A Snape: investigation.
J-H Choi: investigation.
S Bermudez: investigation.
M-N Boivin: investigation and writing—original draft, review, and editing.
N Ferry: investigation and writing—original draft, review, and editing.
J Karamchandani: supervision, investigation, and writing—original draft, review, and editing.
B Nagar: resources, supervision, funding acquisition, investigation, project administration, and writing—original draft, review, and editing.
N Sonenberg: resources, supervision, funding acquisition, project administration, and writing—original draft, review, and editing.

### Conflict of Interest Statement

The authors declare that they have no conflict of interest.

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
