## [Reviewer comments · Life Science Alliance]

Life Science Alliance

PPM1G dephosphorylates eIF4E in control of mRNA translation and cell proliferation.

Peng Wang, Zixian Li, Sung-Hoon Kim, Haijin Xu, Hao Huang, Chutong Yang, Abby Snape, Jung-Hyun Choi, Sara Bermudez, Marie-Noelle Boivin, Nicolas Ferry, Jason Karamchandani, Bhushan Nagar, and Nahum Sonenberg

DOI: <https://doi.org/10.26508/lsa.202402755>

Corresponding author(s): Nahum Sonenberg, McGill University and Bhushan Nagar, McGill University

Review Timeline:

Submission Date:	2024-04-04
Editorial Decision:	2024-05-01
Revision Received:	2024-07-16
Editorial Decision:	2024-07-18
Revision Received:	2024-07-29
Accepted:	2024-07-30

Transaction Report:

May 1, 2024

Re: Life Science Alliance manuscript #LSA-2024-02755-T

Dr. Nahum Sonenberg
McGill University
Department of Biochemistry
Montreal, PQ H3G 1Y6
Canada

Dear Dr. Sonenberg,

Thank you for submitting your manuscript entitled "PPM1G dephosphorylates eIF4E in control of mRNA translation and cell proliferation." to Life Science Alliance. The manuscript was assessed by expert reviewers, whose comments are appended to this letter. We invite you to submit a revised manuscript addressing the Reviewer comments.

Thank you for this interesting contribution to Life Science Alliance. We are looking forward to receiving your revised manuscript.

Sincerely,

B. MANUSCRIPT ORGANIZATION AND FORMATTING:

Reviewer #1 (Comments to the Authors (Required)):

The authors provide clear evidence that in HEK293H cells, the phosphatase PPM1G dephosphorylates the cap binding protein eIF4E. Under the same conditions, PPM1G does not affect the phosphorylation pattern of 4E-BP1. PPM1G is further shown to bind eIF4E through a motif similar to the eIF4E binding motif known in 4E-BP proteins and is able to dephosphorylate eIF4E in vitro. Lastly, whereas overexpression of PPM1G has only a minor impact on global mRNA translation, it strongly inhibits the proliferation of HEK293H and A549 cancer cells, although it remains unclear whether this results from dephosphorylation of eIF4E or other proteins.

The study is important as it refutes previous reports claiming that PPM1G dephosphorylates 4E-BP proteins (PMID: 23814053, 27065332, although only the former is referenced in the present manuscript). The paper also refutes another study showing that PP2A α directly dephosphorylates eIF4E (PMID: 20927323). The data is very clean as far as it goes. However, considering that the present work seeks to revise relatively long-standing concepts in cap-dependent translation, the authors should make more of an attempt to reconcile their new data with the published literature so as not to discredit the previous work unjustly. For example, it appears that a model in which PPM1G controls mRNA translation at the level of both eIF4E and 4E-BP1 cannot be readily dismissed solely based on the current data.

1. The authors contradict the demonstration by Liu et al. that overexpression of PPM1G affects the phosphorylation of 4E-BP1, claiming that "PPM1G did not change the ratio among the isoforms" in the Liu et al. study (P7). However, Fig. 2A of Liu et al. shows that 4E-BP1 isoforms collapse to what would appear to be the alpha isoform in HCT116 cells ectopically expressing FLAG-PPM1G. This raises the possibility that the role of PPM1G may be cell type-dependent. The authors should therefore substantiate the role of PPM1G in dephosphorylating eIF4E (but not 4E-BP1) in additional human and non-human cell lines.
2. The in vitro data in Fig. 2C and D, showing that purified PPM1G dephosphorylates eIF4E look very clear. However, these endpoint data with a single concentration of enzyme and substrate at a single time point do not appropriately assess enzyme kinetics (V_{max} and K_m) that would allow a judgement of whether the data shown is relevant to the in vivo concentrations. The in vitro experiments should also include phosphorylated 4E-BP1 as substrate for comparison to solidify the claim that eIF4E is not a substrate of PPM1G.
3. On P6, it is claimed that "overexpression of PPM1G hardly affected the polysome profile in HEK293H cells (Fig. 4B), indicating that PPM1G-phospho-eIF4E pathway plays a minor role in global mRNA translation". This statement is not fully warranted by the data as the 80S peak is clearly increasing upon PPM1G overexpression, whereas the polysomes appear to decrease. These changes should be quantified in multiple replicates as monosome to polysome ratios with variance and statistics to determine the significance of these changes.
4. As it is speculated in the Discussion (P8) that PPM1G serves to regulate "the translation of a subset of mRNA but not the overall protein synthesis", an attempt should be made to identify that subset. As the authors are surely aware, this can be done with relative ease by sequencing polysomal mRNA in PPM1G overexpressing and depleted cells. It would be quite revealing to compare this subset with the known subset of eIF4E target mRNAs.

Referee Cross Comments:

Regarding Referee 2's statement that the authors "perform experiments showing that PP2A α indirectly regulates eIF4E phosphorylation by targeting its kinases upstream (namely MNKs)", I wish to add that causality is not established here. The data merely shows a correlation between PP2A α and MNKs. Whether this has any role in eIF4E phosphorylation remains unproven.

I concur with the concerns of referee 3, especially with respect to the in vitro phosphatase assay.

Reviewer #2 (Comments to the Authors (Required)):

In this manuscript, Wang et al addressed a key question in the translation field, i.e., what is the phosphatase that dephosphorylates the cap-binding protein eIF4E? This is quite relevant, as eIF4E phosphorylation promotes cell proliferation and is involved in cancer progression. In 2010 a paper by Li et al. reported that PP2AC α is involved in eIF4E dephosphorylation.

Wang et al. demonstrate that the protein phosphatase PPM1G physically interacts with eIF4E and dephosphorylates it. In addition, they perform experiments showing that PP2AC α indirectly regulates eIF4E phosphorylation by targeting its kinases upstream (namely MNKs). This is a remarkable finding, as for decades the identity of eIF4E phosphatase had remained elusive. In this manuscript, the authors also find the eIF4E-interaction motif located in PPM1G, which is quite similar to the one present in other eIF4E-binding proteins. The importance of PPM1G activity in cell biology was also addressed by Wang et al. They show that PPM1G inhibits cell proliferation by targeting phospho-eIF4E-dependent mRNA translation.

I strongly recommend its publication.

Minor points:

In Fig. 4B, the authors perform a polysome profile showing that, upon eIF4E dephosphorylation, no significant changes were observed, suggesting a minor role of it in global translation. Instead, the translation of specific mRNAs could be affected. Could the authors perform either *in vivo* or *in vitro* translation experiments showing that the synthesis of some specific mRNAs are indeed affected by eIF4E dephosphorylation by PPM1G?

Reviewer #3 (Comments to the Authors (Required)):

Comments to the authors

As the major cap-binding protein in eukaryotes, eIF4E is essential for the translation of the majority of mRNAs. Phosphorylation of Ser209 by MNK kinases increases the translation of mRNAs that encode for proteins involved in cell invasion, proliferation, oncogenesis and inflammatory response. As dysregulated translation is a hallmark of multiple human diseases and syndromes, understanding how cells control the phosphorylation levels of eIF4E is important for the potential development of therapeutic strategies.

Our current knowledge on the phosphatases with impact in the amount of phosphorylated eIF4E in cells remains sparse. In this manuscript, Wang and colleagues investigate the phosphatase that directly reduces the levels of phosphorylated eIF4E in human cells. They identify PPM1G as an eIF4E-binding protein that interacts with the cap-binding protein using a motif similar to the canonical eIF4E-binding motif. In addition, using PPM1G overexpression and depletion assays the authors show that the levels of phosphorylated eIF4E correlate with the amount of PPM1G in cells and that PPM1G regulates cell proliferation in an eIF4E-binding dependent manner. Also, *in vitro* phosphatase assays indicate that PPM1G directly dephosphorylates eIF4E. The work presented is original and clearly presented. The authors are also experienced in eIF4E-dependent regulation of translation.

There are some issues that I would recommend that the authors address prior to publication in order to clarify the regulation of eIF4E phosphorylation status in cells.

Major issues:

1. The binding mode of PPM1G to eIF4E

The authors propose that PPM1G has an eIF4E-binding motif that mediates a direct interaction with eIF4E using bacterially purified proteins. It would be important that the authors clarify the position of this motif in the entire protein. Close inspection of the AlphaFold structural model for PPM1G indicates that the eIF4E-binding motif is embedded in the folded phosphatase domain of the protein. How do the authors conceive that this motif mediates binding to eIF4E without disrupting the phosphatase domain? In particular, the Y85 residue seems to mediate interactions with other residues in this domain.

eIF4E-binding proteins contain usually two motifs (canonical and non-canonical) that mediate the interaction with eIF4E. Have the authors also tested this possibility for PPM1G?

The pull-down assay with the bacterially purified proteins suggests that the interaction between the two proteins is not strong (interaction detected only by western blotting). Have the authors also tested the binding of the PPM1G YLmut in the same assay? How do the authors explain that in cells PPM1G competes with other eIF4E-binding proteins to be able to bind and dephosphorylate eIF4E?

The authors refer in the text that PPM1G interacts cap-free eIF4E. How much cap-free (that means not bound to mRNA) and phosphorylated eIF4E is there in human cells? The blots in the manuscript appear to suggest that cap free and phosphorylated eIF4E is not very abundant. Also, what is the amount of PPM1G in cells relative to eIF4E? How does it find the cap-free eIF4E? Do they co-localize in cells?

2. Phosphatase assay

In this study, the authors show using Flag tagged proteins immunoprecipitated from HEK cell extracts, that PPM1G is able to efficiently dephosphorylate eIF4E. Immunoprecipitated PPM1G samples might contain additional proteins that contribute and/or mediate dephosphorylation of eIF4E. Have the authors been able to reproduce the phosphatase assay with the bacterial purified

proteins? Or have they used a catalytic inactive PPM1G protein to show that the phosphorylation activity is solely dependent on PPM1G?

Minor points:

- a. Although the authors have observed a reduction in eIF4E phosphorylation in multiple experiments, the reproducibility of the assays could be supported by quantification of the eIF4E (or others) phosphorylation levels in the different experiments. The methods/legends also don't account for the number of times each experiment was performed.
- b. All blots miss protein sizes and loading amounts on the gels.
- c. Page 5 - "Mutation of the three residues to alanine in PPM1G..." Shouldn't it read to alanine in eIF4E?

We thank the reviewers for their helpful and constructive criticism. In response to their comments and suggestions we amended the paper as follows:

Reviewer #1 (Comments to the Authors (Required)):

The authors provide clear evidence that in HEK293H cells, the phosphatase PPM1G dephosphorylates the cap binding protein eIF4E. Under the same conditions, PPM1G does not affect the phosphorylation pattern of 4E-BP1. PPM1G is further shown to bind eIF4E through a motif similar to the eIF4E binding motif known in 4E-BP proteins and is able to dephosphorylate eIF4E in vitro. Lastly, whereas overexpression of PPM1G has only a minor impact on global mRNA translation, it strongly inhibits the proliferation of HEK293H and A549 cancer cells, although it remains unclear whether this results from dephosphorylation of eIF4E or other proteins.

The study is important as it refutes previous reports claiming that PPM1G dephosphorylates 4E-BP proteins (PMID: 23814053, 27065332, although only the former is referenced in the present manuscript). The paper also refutes another study showing that PP2A α directly dephosphorylates eIF4E (PMID: 20927323). The data is very clean as far as it goes. However, considering that the present work seeks to revise relatively long-standing concepts in cap-dependent translation, the authors should make more of an attempt to reconcile their new data with the published literature so as not to discredit the previous work unjustly. For example, it appears that a model in which PPM1G controls mRNA translation at the level of both eIF4E and 4E-BP1 cannot be readily dismissed solely based on the current data.

Response to reviewer #1:

We thank reviewer for his/her constructive advice. We revised our discussion to reconcile our results with the published literature as follows: “*It was reported that PPM1G dephosphorylates 4E-BP1 (Liu et al., 2013) but whether it dephosphorylates eIF4E was not examined in the latter study. We demonstrated that PPM1G associates with and dephosphorylates eIF4E but not 4E-BP1 (Figs. 1A-C and F). While this discordance needs further investigations, both studies demonstrated that PPM1G reduced eIF4E dependent mRNA translation. Indeed, compared to GFP overexpression, PPM1G overexpression decreased (~ 30%) the average value of polysomes-to-monosome ratio (Fig. 4B) and reduced ~ 30% expression of Mcl-1 (Fig. 4C) which is a validated phospho-eIF4E target (Furic et al., 2010; Martinez et al., 2015; Wendel et al., 2007).*” (please see the discussion).

1. The authors contradict the demonstration by Liu et al. that overexpression of PPM1G affects the phosphorylation of 4E-BP1, claiming that "PPM1G did not change the ratio among the isoforms" in the Liu et al. study (P7). However, Fig. 2A of Liu et al. shows that 4E-BP1 isoforms collapse to what would appear to be the alpha isoform in HCT116 cells ectopically expressing FLAG-PPM1G. This raises the possibility that the role of PPM1G may be cell type-dependent. The authors should therefore substantiate the role of PPM1G in dephosphorylating eIF4E (but not 4E-BP1) in additional human and non-human cell lines.

Response:

We carefully examined again Fig. 2A in the Liu et al. study. Compared to the negative control (row 4, lane 1), overexpression of PPM1G did not change the ratio of 4E-BP1 isoforms (row 4, lane 3). Upon mTOR inhibition by PP242, only unphosphorylated 4E-BP1 (α isoform) can be observed regardless of PPM1G overexpression (row 4, lane 2 and 4). In addition, Fig. 2A in the Liu et al.'s study shows that PPM1G associates only with the hyper-phosphorylated 4E-BP1 but not the unphosphorylated 4E-BP1 (row 2, lane 3), whereas under mTOR inhibitor treatment, PPM1G strongly associates with the unphosphorylated 4E-BP1 (row 2, lane 4). In response to the reviewer's recommendation not to discredit the published work we removed these arguments from the Discussion.

We demonstrated that PPM1G promotes eIF4E dephosphorylation in both HEK293H and A549 cells (Fig. 1A and Fig. 4E), suggesting a conserved mechanism. We would examine our model in other species when cDNAs of PPM1Gs of other species are available.

2. The in vitro data in Fig. 2C and D, showing that purified PPM1G dephosphorylates eIF4E look very clear. However, these endpoint data with a single concentration of enzyme and substrate at a single time point do not appropriately assess enzyme kinetics (V_{max} and K_m) that would allow a judgement of whether the data shown is relevant to the in vivo concentrations. The in vitro experiments should also include phosphorylated 4E-BP1 as substrate for comparison to solidify the claim that eIF4E is not a substrate of PPM1G.

Response:

In Fig. 2C and 2D, we focused on investigating whether PPM1G needs its eIF4E binding sites to efficiently dephosphorylate eIF4E. Therefore, we only used eIF4E but not 4E-BP1 as substrate and we have shown that PPM1G dephosphorylates eIF4E but not 4E-BP1 in Fig. 1F. The authors on this project left due to the lack of research funding. We thus were not able to study the eIF4E dephosphorylation kinetics by PPM1G.

Consequently, we added to the text a note concerning this limitation as follows: “*The study of dephosphorylation kinetics of eIF4E by PPM1G and further investigations of the association/dissociation kinetics of eIF4E with PPM1G, MNKs, 4E-BPs and eIF4G under physiological and pathological conditions are needed for a full understanding of the regulatory mechanism of eIF4E phosphorylation.*”

3. On P6, it is claimed that "overexpression of PPM1G hardly affected the polysome profile in HEK293H cells (Fig. 4B), indicating that PPM1G-phospho-eIF4E pathway plays a minor role in global mRNA translation". This statement is not fully warranted by the data as the 80S peak is clearly increasing upon PPM1G overexpression, whereas the polysomes appear to decrease. These changes should be quantified in multiple replicates as monosome to polysome ratios with variance and statistics to determine the significance of these changes.

Response:

We agree with reviewer, and consequently added the polysomes to monosome ratios with variances and statistics for each polysomes profile graph (Fig. 4B). We found that compared to GFP and PPM1G-YL_{mut} overexpression, PPM1G overexpression insignificantly reduced (~ 30%) the average value of polysomes-monomer ratio (Fig. 4B).

4. As it is speculated in the Discussion (P8) that PPM1G serves to regulate "the translation of a subset of mRNA but not the overall protein synthesis", an attempt should be made to identify that subset. As the authors are surely aware, this can be done with relative ease by sequencing polysomal mRNA in PPM1G overexpressing and depleted cells. It would be quite revealing to compare this subset with the know subset of eIF4E target mRNAs.

Response:

As reviewer suggested, we added a result demonstrating that PPM1G, but not the eIF4E binding mutant of PPM1G, reduced ~ 30% expression of Mcl-1 which is a validated phospho-eIF4E target (Fig. 4C) (Furic *et al*, 2010; Martinez *et al*, 2015; Robichaud *et al*, 2018). We wish to identify the subset of mRNAs in the future.

Referee Cross Comments:

1. Regarding Referee 2's statement that the authors "perform experiments showing that PP2AC α indirectly regulates eIF4E phosphorylation by targeting its kinases upstream (namely MNKs)", I wish to add that causality is not established here. The data merely shows a correlation between PP2AC α

and MNKs. Whether this has any role in eIF4E phosphorylation remains unproven.

Response:

In accordance with the published data (Silverstein *et al*, 2002; Yu *et al*, 2004), we also demonstrated that PP2AC α promoted the dephosphorylation of ERK(MAPK) and MNK2 (Fig. S1E and F), suggesting that PP2AC α may regulate eIF4E phosphorylation by targeting its upstream kinases since MNKs are the only kinase of eIF4E(Ueda *et al*, 2004). We also added a result showing that PP2AC α did not associate with eIF4E (Fig S1G), indicating that PP2AC α does not directly dephosphorylate eIF4E.

2. I concur with the concerns of referee 3, especially with respect to the in vitro phosphatase assay.

Reviewer 3's major issue 2, Phosphatase assay:

In this study, the authors show using Flag tagged proteins immunoprecipitated from HEK cell extracts, that PPM1G is able to efficiently dephosphorylate eIF4E.

Immunoprecipitated PPM1G samples might contain additional proteins that contribute and/or mediate dephosphorylation of eIF4E. Have the authors been able to reproduce the phosphatase assay with the bacterial purified proteins? Or have they used a catalytic inactive PPM1G protein to show that the phosphorylation activity is solely dependent on PPM1G?

Response:

Relative to our purified PPM1G, the co-purified proteins are much less abundant, and they also present in the GFP purified products (Fig. 2C). We think that it's less possible that an additional co-purified protein specifically dephosphorylated eIF4E but not 4E-BP1(Fig. 1F). We added a result demonstrating that catalytically inactive PPM1G mutant (PPM1G-D496A) (Murray *et al*, 1999) did not dephosphorylate eIF4E in an in vitro phosphatase assay (Fig. S1D). Moreover, we found that without Mn²⁺, a metal ion required for PPM1G activity (Travis & Welsh, 1997), PPM1G did not dephosphorylate eIF4E (result will be added in the manuscript on the request). All together, our phosphatase assays results indicate that PPM1G dephosphorylated eIF4E in the test tube.

Reviewer #2 (Comments to the Authors (Required)):

In this manuscript, Wang *et al* addressed a key question in the translation field, i.e., what is the phosphatase that dephosphorylates the cap-binding protein eIF4E? This is quite relevant, as eIF4E phosphorylation promotes cell proliferation and is involved in cancer

progression. In 2010 a paper by Li et al. reported that PP2AC α is involved in eIF4E dephosphorylation.

Wang et al. demonstrate that the protein phosphatase PPM1G physically interacts with eIF4E and dephosphorylates it. In addition, they perform experiments showing that PP2AC α indirectly regulates eIF4E phosphorylation by targeting its kinases upstream (namely MNKs). This is a remarkable finding, as for decades the identity of eIF4E phosphatase had remained elusive. In this manuscript, the authors also find the eIF4E-interaction motif located in PPM1G, which is quite similar to the one present in other eIF4E-binding proteins. The importance of PPM1G activity in cell biology was also addressed by Wang et al. They show that PPM1G inhibits cell proliferation by targeting phospho-eIF4E-dependent mRNA translation.

I strongly recommend its publication.

Response to reviewer #2:

We are very thankful for the support of the reviewer. Please find our responses to the questions below.

Minor points:

In Fig. 4B, the authors perform a polysome profile showing that, upon eIF4E dephosphorylation, no significant changes were observed, suggesting a minor role of it in global translation. Instead, the translation of specific mRNAs could be affected.

Could the authors perform either in vivo or in vitro translation experiments showing that the synthesis of some specific mRNAs are indeed affected by eIF4E dephosphorylation by PPM1G?

Response:

We thank reviewer for the very helpful suggestions. We examined the expression of Mcl-1 whose mRNA translation efficiency is phospho-eIF4E dependent (Furic *et al.*, 2010; Martinez *et al.*, 2015; Robichaud *et al.*, 2018) and found that PPM1G but not PPM1G-YL_{mut} reduced ~30% Mcl-1 expression. Currently, all authors on this project have already left due to the lack of funding and we would further identify the subset of mRNAs in the future when a funding for this project is available.

Reviewer #3 (Comments to the Authors (Required)):

Comments to the authors

As the major cap-binding protein in eukaryotes, eIF4E is essential for the translation of the majority of mRNAs. Phosphorylation of Ser209 by MNK kinases increases the translation of mRNAs that encode for proteins involved in cell invasion, proliferation, oncogenesis and inflammatory response. As dysregulated translation is a hallmark of multiple human diseases and syndromes, understanding how cells control the phosphorylation levels of eIF4E is important for the potential development of therapeutic strategies.

Our current knowledge on the phosphatases with impact in the amount of phosphorylated eIF4E in cells remains sparse. In this manuscript, Wang and colleagues investigate the phosphatase that directly reduces the levels of phosphorylated eIF4E in human cells. They identify PPM1G as an eIF4E-binding protein that interacts with the cap-binding protein using a motif similar to the canonical eIF4E-binding motif. In addition, using PPM1G overexpression and depletion assays the authors show that the levels of phosphorylated eIF4E correlate with the amount of PPMG1 in cells and that PPMG1 regulates cell proliferation in an eIF4E-binding dependent manner. Also, in vitro phosphatase assays indicate that PPM1G directly phosphorylates eIF4E.

The work presented is original and clearly presented. The authors are also experienced in eIF4E-dependent regulation of translation.

There are some issues that I would recommend that the authors address prior to publication in order to clarify the regulation of eIF4E phosphorylation status in cells.

Major issues:

1. The binding mode of PPMG1 to eIF4E

The authors propose that PPM1G has an eIF4E-binding motif that mediates a direct interaction with eIF4E using bacterially purified proteins. It would be important that the authors clarify the position of this motif in the entire protein. Close inspection of the AlphaFold structural model for PPM1G indicates that the eIF4E-binding motif is embedded in the folded phosphatase domain of the protein. How do the authors conceive that this motif mediates binding to eIF4E without disrupting the phosphatase domain? In particular, the Y85 residue seems to mediate interactions with other residues in this domain.

Response:

We thank reviewer for the AlphaFold structural model for PPM1G. Using AlphaFold, we did find a similar model described by reviewer. However, it is difficult to speculate without a structure of the complex, but the YxxxxL motif is at the periphery of the fold, conspicuously on a very short 1.5 turn helix that could possibly flip out to engage eIF4E without unfolding the rest of the PPM1G domain. Notably the Y and L of the motif maintain close proximity when bound to a target or not (please see the images below). Moreover, some posttranslational modifications of PPM1G might adapt its 4E-BM for eIF4E binding. For example, CYFIP1's 4E-BM undergoes an orientation change to be more accessible for eIF4E (Di Marino *et al*, 2015). We decided not to discuss this predicted structure in this study, but we will keep this in mind and would investigate further in the future.

[Figure removed by editorial staff per authors' request]

Major issues:

eIF4E-binding proteins contain usually two motifs (canonical and non-canonical) that mediate the interaction with eIF4E. Have the authors also tested this possibility for PPM1G?

Response:

The non-canonical 4E-BM is crucial for the competition between eIF4G and 4E-BPs for eIF4E binding but poorly conserved in other eIF4E binding proteins (Peter *et al*, 2015; Romagnoli *et al*, 2021). We did not find the non-canonical 4E-BM in PPM1G (discussed in p8), which might explain why PPM1G is less competitive than 4E-BP1 for eIF4E binding (Fig. 3B).

Major issues:

The pulldown assay with the bacterially purified proteins suggests that the interaction between the two proteins is not strong (interaction detected only by western blotting). Have the authors also tested the binding of the PPM1G YLmut in the same assay?

Response:

We agree with reviewer that the interaction between bacterially purified PPM1G and eIF4E (Fig. 1E) is not strong. It is possible that posttranslational modifications of PPM1G in human cells are required for stabilizing the PPM1G-eIF4E interaction. We added a result demonstrating that HIS-PPM1G, but not HIS-PPM1G-YL_{mut}, was pulled down with GST-eIF4E (Fig. S2), indicating that the 4E-BM is required for PPM1G to directly bind to eIF4E.

Major issues:

How do the authors explain that in cells PPM1G competes with other eIF4E-binding proteins to be able to bind and dephosphorylate eIF4E?

Response:

This is a very fundamental question. There are eleven identified human eIF4E-binding proteins which contain the canonical 4E-BM with or without additional non-canonical 4E-BM (Romagnoli *et al.*, 2021). How they compete for eIF4E-binding is not known.

Major issues:

The authors refer in the text that PPM1G interacts cap-free eIF4E. How much cap-free (that means not bound to mRNA) and phosphorylated eIF4E is there in human cells? The blots in the manuscript appear to suggest that cap free and phosphorylated eIF4E is not very abundant. Also, what is the amount of PPM1G in cells relative to eIF4E? How does it find the cap-free eIF4E? Do they co-localize in cells?

Response:

We thank reviewer for his excellent questions. It is thought that after each round of translation initiation, eIF4E and other translation initiation factors are released. We think that released eIF4E may undergo the phosphorylation/dephosphorylation regulation. However, we did not find a way to quantify the phosphorylated cap-free eIF4E in the cells and we cannot tell either the relative quantity of endogenous PPM1G and eIF4E as both proteins were poorly immunoprecipitated. We hope that these interesting questions will be resolved in the future. eIF4E is mainly cytoplasmic and partially (~30%) nuclear (Lejbkiewicz *et al.*, 1992) while PPM1G is predominantly nuclear with sparse cytoplasmic localization (Kumar *et al.*, 2019). Moreover, it was reported that PPM1G may be shuttling between the nucleus and the cytoplasm (Liu *et al.*, 2013) and may target its cytoplasmic

substrate by associating with B56 δ (Kumar *et al.*, 2019). PPM1G thus might dephosphorylate cap-free eIF4E both in the cytoplasm and in the nucleus.

Major issues:

2. Phosphatase assay

In this study, the authors show using Flag tagged proteins immunoprecipitated from HEK cell extracts, that PPM1G is able to efficiently dephosphorylate eIF4E.

Immunoprecipitated PPM1G samples might contain additional proteins that contribute and/or mediate dephosphorylation of eIF4E. Have the authors been able to reproduce the phosphatase assay with the bacterial purified proteins? Or have they used a catalytic inactive PPM1G protein to show that the phosphorylation activity is solely dependent on PPM1G?

Response:

We thank the reviewer for the excellent questions. Relative to our purified PPM1G, the co-purified proteins are much less abundant, and they are also present in the GFP purified products (Fig. 2C). We think that it's less possible that an additional protein specifically dephosphorylated eIF4E but not 4E-BP1(Fig. 1F). We added a result demonstrating that catalytically inactive PPM1G mutant (PPM1G-D496A)(Murray *et al.*, 1999) did not dephosphorylate eIF4E in an *in vitro* phosphatase assay (Fig. S1D). Moreover, without Mn²⁺, metal ion required for PPM1G activity (Travis & Welsh, 1997), PPM1G did not dephosphorylate eIF4E (result will be added in the manuscript on the request). All together, our phosphatase assays results indicate that PPM1G dephosphorylated eIF4E in the test tube (Fig. 1F, S1D and 2D).

Minor points:

a. Although the authors have observed a reduction in eIF4E phosphorylation in multiple experiments, the reproducibility of the assays could be supported by quantification of the eIF4E (or others) phosphorylation levels in the different experiments. The methods/legends also don't account for the number of times each experiment was performed.

Response:

We quantified the eIF4E dephosphorylation by PPM1G in Fig. 2B and in Fig. 4A. We did not quantify some experiments which clearly showed that PPM1G promoted the dephosphorylation of eIF4E, but we followed the reviewer's suggestion and stated in Methods section that all the results in this study were reproduced at least three times.

b. All blots miss protein sizes and loading amounts on the gels.

Response:

To not overload some figures, we provided the molecular weights in a supplementary material, and we indicated the loading amounts as well.

c. Page 5 - "Mutation of the three residues to alanine in PPM1G..." Shouldn't it read to alanine in eIF4E?

Response:

We thank reviewer and we corrected it in our manuscript.

Di Marino D, Chillemi G, De Rubeis S, Tramontano A, Achsel T, Bagni C (2015) MD and Docking Studies Reveal That the Functional Switch of CYFIP1 is Mediated by a Butterfly-like Motion. *J Chem Theory Comput* 11: 3401-3410

Furic L, Rong L, Larsson O, Koumakpayi IH, Yoshida K, Brueschke A, Petroulakis E, Robichaud N, Pollak M, Gaboury LA *et al* (2010) eIF4E phosphorylation promotes tumorigenesis and is associated with prostate cancer progression. *Proceedings of the National Academy of Sciences of the United States of America* 107: 14134-14139

Kumar P, Tathe P, Chaudhary N, Maddika S (2019) PPM1G forms a PPP-type phosphatase holoenzyme with B56delta that maintains adherens junction integrity. *EMBO reports*: e46965

Lejbkowitz F, Goyer C, Darveau A, Neron S, Lemieux R, Sonenberg N (1992) A fraction of the mRNA 5' cap-binding protein, eukaryotic initiation factor 4E, localizes to the nucleus. *Proceedings of the National Academy of Sciences of the United States of America* 89: 9612-9616

Liu J, Stevens PD, Eshleman NE, Gao T (2013) Protein phosphatase PPM1G regulates protein translation and cell growth by dephosphorylating 4E binding protein 1 (4E-BP1). *The Journal of biological chemistry* 288: 23225-23233

Martinez A, Sese M, Losa JH, Robichaud N, Sonenberg N, Aasen T, Ramon YCS (2015) Phosphorylation of eIF4E Confers Resistance to Cellular Stress and DNA-Damaging Agents through an Interaction with 4E-T: A Rationale for Novel Therapeutic Approaches. *PloS one* 10: e0123352

Murray MV, Kobayashi R, Krainer AR (1999) The type 2C Ser/Thr phosphatase PP2Cgamma is a pre-mRNA splicing factor. *Genes & development* 13: 87-97

Peter D, Igreja C, Weber R, Wohlbold L, Weiler C, Ebertsch L, Weichenrieder O, Izaurralde E (2015) Molecular architecture of 4E-BP translational inhibitors bound to eIF4E. *Molecular cell* 57: 1074-1087

Robichaud N, Hsu BE, Istomine R, Alvarez F, Blagih J, Ma EH, Morales SV, Dai DL, Li G, Souleimanova M *et al* (2018) Translational control in the tumor microenvironment promotes lung metastasis: Phosphorylation of eIF4E in neutrophils. *Proceedings of the National Academy of Sciences of the United States of America* 115: E2202-E2209

Romagnoli A, D'Agostino M, Ardiccioni C, Maracci C, Motta S, La Teana A, Di Marino D (2021) Control of the eIF4E activity: structural insights and pharmacological implications. *Cellular and molecular life sciences : CMLS*

Silverstein AM, Barrow CA, Davis AJ, Mumby MC (2002) Actions of PP2A on the MAP kinase pathway and apoptosis are mediated by distinct regulatory subunits. *Proceedings of the National Academy of Sciences of the United States of America* 99: 4221-4226

Travis SM, Welsh MJ (1997) PP2C gamma: a human protein phosphatase with a unique acidic domain. *FEBS letters* 412: 415-419

Ueda T, Watanabe-Fukunaga R, Fukuyama H, Nagata S, Fukunaga R (2004) Mnk2 and Mnk1 are essential for constitutive and inducible phosphorylation of eukaryotic initiation factor 4E but not for cell growth or development. *Molecular and cellular biology* 24: 6539-6549

Yu LG, Packman LC, Weldon M, Hamlett J, Rhodes JM (2004) Protein phosphatase 2A, a negative regulator of the ERK signaling pathway, is activated by tyrosine phosphorylation of putative HLA class II-associated protein I (PHAPI)/pp32 in response to the antiproliferative lectin, jacalin. *The Journal of biological chemistry* 279: 41377-41383

July 18, 2024

RE: Life Science Alliance Manuscript #LSA-2024-02755-TRR

Dr. Nahum Sonenberg
McGill
Department of Biochemistry
Dept. of Biochemistry
McGill University
Montreal, PQ H3G 1Y6
Canada

Dear Dr. Sonenberg,

Thank you for submitting your revised manuscript entitled "PPM1G dephosphorylates eIF4E in control of mRNA translation and cell proliferation.". We would be happy to publish your paper in Life Science Alliance pending final revisions necessary to meet our formatting guidelines.

- please be sure that the authorship listing and order is correct, and that they match in our system and in the manuscript
- please add the Twitter handle of your host institute/organization as well as your own or/and one of the authors in our system
- please be sure that all authors are mentioned in the Author Contributions section
- the contributions selected for Marie-Noelle Boivin, Nicolas Ferry, and Jason Karamchandani do not qualify them for authorship. Please either update the contributions in our system and the Author Contributions section of the manuscript or let us know if the authors need to be removed (and added eventually to the Acknowledgments section)
- please add your main and supplementary figure legends to the main manuscript text after the references section
- please make sure the manuscript sections are aligned following LSA's formatting guidelines: please separate the Figure legends and Supplemental Figure legends into separate sections
- Figure S2 has only one panel; please remove label A from its legends and call out in the manuscript text

LSA now encourages authors to provide a 30-60 second video where the study is briefly explained. We will use these videos on social media to promote the published paper and the presenting author (for examples, see <https://docs.google.com/document/d/1-UWCfbE4pGcDdcgzcmiuJl2XMBJnxKYeqRvLLrLS08s/edit?usp=sharing>). Corresponding or first-authors are welcome to submit the video. Please submit only one video per manuscript. The video can be emailed to contact@life-science-alliance.org

A. FINAL FILES:

B. MANUSCRIPT ORGANIZATION AND FORMATTING:

Sincerely,

July 30, 2024

RE: Life Science Alliance Manuscript #LSA-2024-02755-TRRR

Dr. Nahum Sonenberg
McGill University
Department of Biochemistry
1160 Pine Avenue West
Room 615
Montreal, PQ H3A 1A3
Canada

Dear Dr. Sonenberg,

Thank you for submitting your Research Article entitled "PPM1G dephosphorylates eIF4E in control of mRNA translation and cell proliferation.". It is a pleasure to let you know that your manuscript is now accepted for publication in Life Science Alliance. Congratulations on this interesting work.

DISTRIBUTION OF MATERIALS:

Again, congratulations on a very nice paper. I hope you found the review process to be constructive and are pleased with how the manuscript was handled editorially. We look forward to future exciting submissions from your lab.

Sincerely,
